# The role of social norms on adolescent family planning in rural Kilifi county, Kenya

Shaon Lahiri[1,2,3]*, Jeffrey Bingenheimer[1], Erica Sedlander[1,4], Wolfgang Munar[5], Rajiv Rimal[6]

**1** Department of Prevention and Community Health, Milken Institute School of Public Health, The George Washington University, Washington, DC, United States of America, **2** Philosophy, Politics and Economics Program, University of Pennsylvania, Philadelphia, Pennsylvania, United States of America, **3** Center for Social Norms and Behavioral Dynamics, University of Pennsylvania, Philadelphia, Pennsylvania, United States of America, **4** Institute for Health and Aging, University of California, San Francisco, San Francisco, California, United States of America, **5** Department of Global Health, Milken Institute School of Public Health, The George Washington University, Washington, DC, United States of America, **6** Department of Health, Behavior, and Society, Bloomberg School of Public Health, The Johns Hopkins University, Baltimore, Maryland, United States of America

\* shaonl@sas.upenn.edu

## Abstract

**Data Availability Statement:** All relevant data are available at Open Science Framework. The Role of Social Norms on Adolescent Family Planning in Kilifi County, Kenya. Available at: https://doi.org/10.17605/OSF.IO/AYC27.

### Purpose

Despite Kenya's encouraging progress in increasing access to modern contraception among youth, several barriers remain preventing large-scale efforts to reduce demand-side unmet need for family planning. Shifting social norms around the use and acceptability of modern contraception may represent a potent target for future interventions. However, the structure of normative influence on individual modern contraceptive use among youth needs to be determined. Therefore, our aim was to estimate the influence of individual and group-level normative influence on modern contraceptive use among adolescents from two villages in rural Kenya.

### Methods

Trained enumerators collected data from individuals aged 15–24 who provided oral informed consent, or parental informed consent, in two villages in rural Kilifi county. Participants completed a questionnaire related to modern contraceptive use and were asked to nominate one to five people (referents) with whom they spend free time. The enumerators photographed each individual who nominated at least one referent using Android phones and matched them with their nominated referents. Using this social network data, we estimated group-level normative influence by taking an average of referents' modern contraception use. We then explored associations between descriptive norms, injunctive norms, and network modern contraceptive use on individual modern contraceptive use, controlling for known confounders using logistic regression models. We also conducted sensitivity analyses to test a pattern of differential referent influence on individual modern contraceptive use.

**Funding:** This work was supported, in whole or in part, by the Bill & Melinda Gates Foundation [grant: OPP1135005]. Under the grant conditions of the Foundation, a Creative Commons Attribution 4.0 Generic License has already been assigned to the Author Accepted Manuscript version that might arise from this submission. The funders had no role in study design, data collection and analysis, decision to publish, or preparation of the manuscript.

**Competing interests:** The authors have declared that no competing interests exist.

## Results

There was a positive association between pro-modern contraception descriptive and injunctive norms and individual modern contraception use (adjusted Odds Ratio (aOR) = 1.29, 95% confidence interval (CI) = 1.05–1.6, and aOR = 1.31, CI = 1.06–1.62, respectively). Network modern contraceptive use was associated with individual use in the bivariate model (aOR = 2.57, CI = 1.6–4.12), but not in the multivariable model (aOR = 1.67, CI = 0.98–2.87). When stratified by sex and marital status, network modern contraceptive use was associated with individual modern contraceptive use among female participants (aOR = 2.9, CI = 1.31–6.42), and unmarried female participants (aOR = 5.26, CI = 1.34–20.69), but not among males. No interactive effects between norms variables were detected. Sensitivity analyses with a different estimate of network modern contraceptive use showed similar results.

## Conclusions

Social norms are multilevel phenomena that influence youth modern contraceptive use, especially among young women in rural Kenya. Unmarried women with modern contraceptive users in their social network may feel less stigma to use contraception themselves. This may reflect gendered differences in norms and social influence effects for modern contraceptive use. Future research should investigate group-level normative influence in relation to family planning behaviors.

## Introduction

Kenya has exceeded its 2020 target of 58% of married women using modern family planning methods [1]. Recent results from the nationally representative Performance Monitoring and Accountability 2020 (PMA2020) survey suggest that approximately 61% of women aged 15–49 who were married or in a union reported using modern contraception (MC) [2]. Despite this encouraging progress, barriers to modern contraceptive uptake remain, particularly among youth. In 2021, among married women aged 15–19, the estimated modern contraceptive prevalence was 48.2%, and for married women aged 20–24, it was 58.4% [3]. Additionally, in the same time period, 59.1% of women aged 15–19, and 41.9% of women aged 20–24, with at least one birth wanted to have their pregnancy later than they did [3].

The prevalence of modern contraceptive methods among youth in Kenya is especially important given that the majority of Kenya's population comprise young people. According to the 2019 Census, an estimated 39% of the population is under 15, and 58% are under 24 [4]. Additionally, according to the 2014 Kenya Demographic and Health survey (KDHS), which was the last available, nearly one out of every five adolescent girls aged 15–19 reported being pregnant or giving birth [5], meaning that a large share of Kenya's demographic pyramid comprises young people who are parents.

While teen pregnancy has been a perennial concern in Kenya, it appears to have been exacerbated by the COVID-19 pandemic. During three months of lockdown in 2020, an estimated 152,000 Kenyan teenage girls became pregnant, representing a 40% increase in the monthly average [6]. This has drastic implications on school dropout and transfer rates. An analysis of 12 secondary schools in Western Kenya from 2018–2021 found that girls experiencing COVID-19 containment measures had twice the risk of becoming pregnant, three times the

risk of dropping out of school, and more than three times the risk of transferring schools before an examination period, compared to girls who graduated secondary school before the pandemic [7]. In this way, the advent of COVID-19 may derail gains in sexual and reproductive health outcomes among youth and worsen existing challenges, particularly among young women.

Kenya's National Adolescent Sexual and Reproductive Health Policy recognizes that many adolescent pregnancies are unintended and can result from sexual abuse, child marriage, or coerced sex, strengthening the need to improve family planning efforts for adolescents [8]. However, as pointed out by Psaki [9], policies and laws often do not sufficiently address the confluence of factors that contribute to early marriage and adolescent pregnancy. There is also significant geographic variation in the unmet need for family planning among all women aged 15–49, with the highest unmet need in West Pokot (19.6%), and the lowest in Nairobi and Kiambu (6.2%) [3]. Additionally, among urban women of reproductive age, teenagers and those from lower-income households have a greater odds of having an unmet need for family planning [10]. Thus, despite gains in modern contraceptive use in Kenya, there still exist significant blind spots, particularly across settings and demographic factors.

Social factors such as stigma against unmarried youth using contraceptives, fears of infertility and side effects, and educators reporting difficulties discussing sexuality with adolescents also hinder uptake of family planning methods [11–14]. These factors also tend to have stark gendered and marital dimensions. Sexual activity before marriage has been described by abortion care providers as immoral [15, 16], and the use of contraception by young women in Kenya has been described as encouraging women's promiscuous behavior and decreasing their sexual arousal leading to extramarital sexual relationships among husbands [12, 17]. In this way, young women in Kenya face immense pressure to remain abstinent until marriage, with punitive social sanctions for deviation from entrenched social norms [18, 19].

Social norms may therefore be an important target for improving family planning outcomes in the country. Meta-analytic results from mostly high-income countries suggest that social norms can strongly influence adolescent sexual behavior, with sociocultural context significantly moderating the norms-behavior relationship [20]. In the Kenyan context, social norms have been a recognized target of family planning programs [21, 22], although relatively little empirical research has explored granular associations between social norms and family planning behaviors in the country.

Part of the difficultly in studying social norms around family planning lies in accurately measuring them. To capture the influence of social normative perceptions related to specific behaviors (e.g., modern contraceptive use), researchers typically measure individual perceptions of how common (i.e., descriptive norms) and appropriate (i.e., injunctive norms) a behavior is within a particular reference group [23]. A reference group comprises the individuals whose beliefs, opinions, and behaviors matter to an individual's decision-making and behavior [24]. Some referents may have greater influence in the group than others for particular behaviors, and referents are thought to influence group members' normative perceptions through social interaction and observation [25]. In this way, determining which types of individuals (e.g. family, friends, others) most often comprise young people's reference groups for family planning behaviors can aid in the development of targeted programs and policies.

However, social norms do not exist solely as individual perceptions. They can also exist as group-level social influence, manifested in the actual (versus perceived) prevalence of a behavior within a reference group. This manifestation of social norms is termed "collective norms" in the Theory of Normative Social Behavior [26], and has been found to influence behavior independent of perceived norms. Descriptive, injunctive, and collective norms are also thought

to interact with one another to influence behavior [27], though evidence for these interaction effects in the family planning literature is scant.

Some studies have found a significant relationship between collective norms and individual family planning behaviors [28, 29]. In these studies, investigators estimated collective norms by aggregating individual behaviors to the enumeration area or higher-level clusters using existing DHS data. However, given the variation in social norms within geographical clusters, estimating collective norms within an individual's specified reference group may provide more accurate estimates of social influence effects on individual norms perceptions and behavior than a more geographically aggregate approach [25]. A study by Shakya et al. [30] was unique in that it assessed the influence of proximal reference group members' normative perceptions on adolescent childbirth, in addition to village-level injunctive norms and behavioral prevalence. To our knowledge, the relationship between collective norms, operationalized as behavioral prevalence within a proximal social network, and individual behavior has not yet been applied in the adolescent family planning literature in sub-Saharan Africa.

To address these gaps, we estimated the direct and interactional influence of collective norms (operationalized as *network MC use)* and perceived norms on individual MC use among a sample of adolescents aged 15–24 in two Kenyan villages. We hypothesized that network MC use, descriptive norms, and injunctive norms would all be positively associated with individual MC use. Additionally, we also believed that there would be significant interactive effects between descriptive norms, injunctive norms, and network MC use as they relate to individual MC use. These hypotheses stem from the expanded Theory of Normative Social Behavior which suggests that the three types of norms may operate differently in influencing male and female adolescent behavior, and that interactions between normative variables can be used to understand the effect of different normative scenarios on behavior [27]. As an exploratory aim, we wanted to know the most common reference group identified for male and female participants. Given the gendered dimension of MC use in this context, and the difficulties faced in MC access and use among unmarried women in particular, we also stratified our analyses by sex and marital status to explore potential subgroup effects.

## Methods

### Ethics statement

Consent was obtained verbally for this project, as witnessed by research team staff. Youth assent and parental consent were obtained in this way, and this was documented in writing. Once informed consent was obtained, photos of residents were taken using Android phones and were then uploaded into Trellis along with the roster data (69 percent of all residents from the roster had their photo taken). Trellis issued unique identifiers for each respondent that were matched to their photos and demographic information. This procedure was approved by the Institutional Review Board of Northwestern University (STU00203585).

### Data and study design

In the present study, we analyzed an adolescent subgroup dataset assembled from a cross-sectional sociocentric survey conducted in 2018 in two villages in Kilifi County, Kenya. The villages were selected on the basis of their reported MC prevalence among women of childbearing age (aged 15–49 years)–one with lower prevalence (10.3%) and one with higher prevalence (44.4%) (hereafter referred to as "Lower MC Use Village" and "Higher MC Use Village", respectively). The purpose of selecting these two villages was to include settings at different levels of fertility transition (i.e. where the decline from high to lower fertility was faster in

one village compared to the other). The Higher MC Use village represented a setting with a relatively more advanced stage of fertility transition, as compared with the Lower MC Use village.

Trained enumerators from a Kenyan research agency used a sociocentric approach and collected data from all residents aged 15 or older. Those aged 18 years or older provided oral informed consent. For individuals who 15 to 17 years, we obtained their informed assent, and parental informed consent. The enumerators photographed each individual (ego) using Android phones and matched them with their nominated referents (alters) using the *Trellis* software [31]. The sample comprised individuals who were married or living with a partner, as well as participants who were unmarried and not cohabiting.

The aim of the data collection was to survey 100% of men and women who were eligible in each of the villages. However, the overall response rate across both villages was 72%, and most of the non-response was due to inability to locate certain residents. The term "medical methods of family planning" was used throughout the survey in place of "modern contraception." Our analysis is restricted to male and female participants aged 15–24 years.

### Measures

The main outcome for this study was current individual MC use, as assessed by a self-reported binary indicator ("Yes/No"). The focal independent variables for this study comprised perceived descriptive and injunctive norms, as well as network MC Use.

To construct the network MC use variable, we first populated MC use for each individual's referents. The referents comprised one to five people whom the individual nominated in response to the question "With whom did you spend a lot of free time in the past year?" This question was selected based on discussions with local data collection partners and pilot tests. Some individuals nominated more referents than others, and the sample was restricted to those who nominated at least one referent. As a result, 32 individuals were dropped from the regression analyses, as they did not nominate any referents. We then took an average across all referents' MC use, excluding the respondent's own MC use, to derive network MC use. To capture the possible differential influence of referent MC use (i.e. negative influence from non-using referents and positive influence from using referents), as well as to allow for parity between those who nominated greater versus fewer referents, we modified the network MC use variable for sensitivity analyses. In the modified version, we imputed a value of 0 if the referent's MC use was missing, -1 if the referent did not use MC, and +1 if the referent used MC.

Perceived descriptive norms and injunctive norms were created by averaging two items for each variable. Response options for these items were on a 5-point Likert scale. For perceived descriptive norms, the items were: 1) "Most people around me use medical methods of family planning for determining when to have a child" and 2) "Most people whose opinions I value use medical methods of family planning for determining if or when to have a child." The response options for these items went from Strongly Agree (1) to Strongly Disagree (5). For perceived injunctive norms, the items were: 1) "Most people important to me will think badly of me if I use medical methods of family planning" and 2) "Most people important to me will reject me if I use medical methods of family planning." The response options for these items were recoded to range from Strongly Disagree (1) to Strongly Agree (5). In this way, higher values for perceived descriptive norms, network MC use, and perceived injunctive norms indicate pro-MC values.

We also selected several sociodemographic variables known to be associated with MC use such as age, sex, education level, religion, marital status, MC beliefs, MC attitudes, and village of residence. For MC beliefs and MC attitudes, we averaged two items for each variable that we felt represented each construct. Variables were coded such that higher values represent pro-

MC responses. For example, response options for the MC belief "Using medical methods of family planning can cause health problems" ranged from Strongly Agree (1) to Strongly Disagree (5). Complete item wording and response options for key variables are available in S1 Table in S1 File.

## Statistical analyses

To understand the relationship between social norms and individual MC use, we constructed several bivariate and multivariable logistic regression models. We computed odds ratios and centered continuous variables at their grand mean in regression models with interaction variables to facilitate interpretation of coefficients. We also stratified our analyses by sex and marital status to explore differences in MC use and social norms by these two dimensions. The choice of these two strata was motivated by previous studies illustrating gender differences in MC use (see for example, Jalang'o et al. [32]), and that MC use during marriage can be subject to intra-household bargaining, among other factors (see for example, Obare et al. [33]). To explore differences in sociodemographic variables between villages, we performed chi-square tests of independence for categorical variables, and t-tests for continuous variables. We used an *a priori* alpha of 0.05 for all significance tests.

We tested five multivariable models: Model (2) has no interaction terms; Model (3) has an interaction between perceived injunctive and descriptive norms; Model (4) has an interaction between network MC use and perceived descriptive norms; Model (5) has an interaction between network MC use and perceived injunctive norms; and Model (6) has all three interaction terms simultaneously. All analyses were performed in R [34] and Stata 15 [35].

## Results

Descriptive statistics and frequencies for the sample are presented in **Table 1**. Overall, there were a total of 763 respondents across both villages, and the majority of the sample were Muslim (55%), female (57%), had completed post-primary school or lower (61%), and had never been married (72%). The mean age was 19 years, and over 95% of the sample had nominated at least one referent. While there were 207 more respondents from the Higher MC Use Village than the Lower MC Use Village, the two villages were broadly comparable on demographic and perceived norms. However, respondents from the Lower MC Use Village were more likely to be Christian, slightly younger, have lower Network MC Use, and have nominated fewer referents compared to the Higher MC Use Village.

Additionally, we computed Pearson correlations between Network MC Use, perceived descriptive norms, and perceived injunctive norms. None of the correlations were significant, and all were less than 0.1.

The prevalence of each type of MC use is presented, by village and overall, in **Table 2**.

The sample was characterized by generally low MC use, with only 29% reporting one or more types of MC use. Overall, the most commonly reported MC type was the male condom (15%). Hormonal methods of MC, comprising oral contraceptive pills, emergency contraceptive pills, and injectables were used by 15.1% of the total sample. Among non-MC users, the vast majority were not using traditional contraceptive methods either (86% of non-MC users).

Frequencies of referent categories by sex are presented in **Table 3.**

Among women, a same-sex relation was most frequently nominated. The five most frequently nominated referent categories among women were 1) a female relative not mentioned in the given list; 2) sister-in-law; 3) mother; 4) female friend; and 5) sister. A similar pattern held for male participants, though they also nominated some female relations. The five most

**Table 1. Descriptive statistics.**

| Variable | Overall[1] (N = 763) | Higher MC Use Village[1] (N = 485) | Lower MC Use Village[1] (N = 278) | p-value[2] |
|---|---|---|---|---|
| **Education** | | | | 0.3 |
| No schooling   Completed | 27 (3.5%) | 16 (3.3%) | 11 (4.0%) | |
| Primary | 381 (50%) | 239 (49%) | 142 (51%) | |
| Post-primary/Vocational/Other | 57 (7.5%) | 41 (8.5%) | 16 (5.8%) | |
| Secondary/'A' level | 257 (34%) | 168 (35%) | 89 (32%) | |
| College/University | 41 (5.4%) | 21 (4.3%) | 20 (7.2%) | |
| **Religion** | | | | <0.001 |
| Roman Catholic | 12 (1.6%) | 3 (0.6%) | 9 (3.2%) | |
| Protestant/Other Christian | 292 (38%) | 128 (26%) | 164 (59%) | |
| Muslim | 418 (55%) | 354 (73%) | 64 (23%) | |
| No Religion | 41 (5.4%) | 0 (0%) | 41 (15%) | |
| **Sex** | | | | 0.8 |
| Male | 325 (43%) | 209 (43%) | 116 (42%) | |
| Female | 438 (57%) | 276 (57%) | 162 (58%) | |
| **Marital Status** | | | | 0.10 |
| Living with a man/woman | 8 (1.0%) | 8 (1.6%) | 0 (0%) | |
| Married (monogamous) | 178 (23%) | 119 (25%) | 59 (21%) | |
| Married (polygamous) | 14 (1.8%) | 11 (2.3%) | 3 (1.1%) | |
| Separated/Divorced/Widowed | 16 (2.1%) | 10 (2.1%) | 6 (2.2%) | |
| Single (never married) | 547 (72%) | 337 (69%) | 210 (76%) | |
| **Age** | 19.08 (2.80) | 19.29 (2.72) | 18.72 (2.92) | 0.008 |
| **Number of Nominated Referents** | | | | <0.001 |
| 0 | 32 (4.2%) | 22 (4.5%) | 10 (3.6%) | |
| 1 | 104 (14%) | 76 (16%) | 28 (10%) | |
| 2 | 216 (28%) | 149 (31%) | 67 (24%) | |
| 3 | 207 (27%) | 138 (28%) | 69 (25%) | |
| 4 | 139 (18%) | 76 (16%) | 63 (23%) | |
| 5 | 65 (8.5%) | 24 (4.9%) | 41 (15%) | |
| **Network MC Use** | 0.31 (0.33) | 0.34 (0.34) | 0.26 (0.31) | <0.001 |
| Missing | 32 | 22 | 10 | |
| **Perceived Descriptive Norms** | 3.59 (0.93) | 3.56 (0.82) | 3.63 (1.09) | 0.4 |
| **Perceived Injunctive Norms** | 3.32 (0.96) | 3.33 (0.91) | 3.30 (1.05) | 0.7 |

[1]n (%); Mean (SD)

[2]Pearson's Chi-squared test; Welch Two Sample t-test

frequently nominated referent categories among male participants were 1) male friend; 2) a male relative not mentioned in the given list; 3) brother; 4) mother; and 5) sister.

Bivariate and multivariable analyses are presented in **Table 4**.

Model (1) in Table 4 presents the bivariate results, which illustrate a positive relationship between network MC use and individual MC use. We also found positive relationships between perceived injunctive and descriptive norms with individual MC use. Female participants were less likely, and married participants were more likely to use MC. Additionally, we found positive relationships between the two MC use beliefs ("MC use leads to health problems" and "MC use leads to lost trust between partners") and individual MC use. The two attitudinal variables ("Sexual activity is ok" and "MC use is ok") were also positively associated with individual MC use. Our hypotheses related to the direct relationships between the norms

**Table 2. Traditional and modern contraceptive use by village.**

| Variable | Overall[1] (N = 763) | Not Married[1] (N = 571) | Married[1] (N = 192) | p-value[2] |
|---|---|---|---|---|
| Male Condom | 111 (15%) | 107 (19%) | 4 (2.1%) | <0.001 |
| Implants | 51 (6.7%) | 13 (2.3%) | 38 (20%) | <0.001 |
| Injectables | 48 (6.3%) | 10 (1.8%) | 38 (20%) | <0.001 |
| Pill | 10 (1.3%) | 5 (0.9%) | 5 (2.6%) | 0.15 |
| Emergency contraception | 8 (1.0%) | 8 (1.4%) | 0 | |
| Female Condom | 1 (0.1%) | 1 (0.2%) | 0 | |
| Female Sterilization | 0 | 0 | 0 | |
| Male Sterilization | 0 | 0 | 0 | |
| IUD & 'the coil' | 0 | 0 | 0 | |
| Lactational Amenorrhea Method (Breastfeeding) | 0 | 0 | 0 | |
| **Type of MC Use** | | | | <0.001 |
| Some MC use | 222 (29%) | 138 (24%) | 84 (44%) | |
| No MC use | 541 (71%) | 433 (76%) | 108 (56%) | |
| **Traditional Contraception Use among non-MC users** | | | | >0.9 |
| No traditional contraceptive use | 190 (86%) | 114 (86%) | 76 (86%) | |
| Traditional contraceptive use | 31 (14%) | 19 (14%) | 12 (14%) | |

[1]n (%)

[2]Pearson's Chi-squared test

**Table 3. Distribution of referent categories for male and female participants.**

| Nominee Category (Female Participants) | N (%) |
|---|---|
| Other female relative | 432 (35.35%) |
| Sister-in-law | 144 (11.78%) |
| Mother | 134 (10.97%) |
| Female friend | 111 (9.08%) |
| Sister | 103 (8.43%) |
| Co-wife | 60 (4.91%) |
| Mother-in-law | 52 (4.26%) |
| Brother | 51 (4.17%) |
| Husband | 30 (2.45%) |
| Other | 105 (8.59%) |
| *Total (Female Participants)* | *1222 (100.00%)* |
| **Nominee Category (Male Participants)** | **N (%)** |
| Male friend | 213 (25.94%) |
| Other male relative | 206 (25.09%) |
| Brother | 142 (17.30%) |
| Mother | 69 (8.40%) |
| Sister | 51 (6.21%) |
| Other female relative | 47 (5.72%) |
| Father | 43 (5.24%) |
| Wife | 12 (1.46%) |
| Grandmother | 12 (1.46%) |
| Other | 26 (3.17%) |
| *Total (Male Participants)* | *821 (100.00%)* |

**Table 4. Bivariate and multivariable regression models.**

| | Individual MC Use (N = 727) | | | | | |
|---|---|---|---|---|---|---|
| | Bivariate OR (95% CI) | Adjusted OR (95% CI) | | | | |
| | (1) | (2) | (3) | (4) | (5) | (6) |
| Network MC Use | 2.57*** (1.60, 4.12) | 1.67(0.98, 2.87) | 1.66(0.97, 2.85) | 1.64(0.95, 2.83) | 1.58(0.9,2.75) | 1.58(0.9,2.76) |
| Perceived Descriptive Norms | 1.28** (1.07, 1.54) | 1.31* (1.06,1.62) | 1.28* (1.03,1.59) | 1.31* (1.06,1.63) | 1.31* (1.05,1.62) | 1.29* (1.03,1.6) |
| Perceived Injunctive Norms | 1.50*** (1.26, 1.80) | 1.29* (1.05,1.6) | 1.23 (0.99,1.53) | 1.29* (1.04,1.59) | 1.29* (1.05,1.6) | 1.24 (1,1.54) |
| Perceived Injunctive Norms x Perceived Descriptive Norms | | | 1.20 (0.97,1.48) | | | 1.18 (0.96,1.46) |
| Network MC Use x Perceived Descriptive Norms | | | | 1.27 (0.66,2.45) | | 1.22 (0.63,2.38) |
| Network MC Use x Perceived Injunctive Norms | | | | | 1.33(0.7,2.51) | 1.21(0.63,2.31) |
| Age | 1.17*** (1.11, 1.25) | 1.10* (1.02,1.19) | 1.11* (1.02,1.2) | 1.10* (1.02,1.19) | 1.10* (1.02,1.19) | 1.11* (1.02,1.2) |
| Married [Ref: Unmarried] | 2.45*** (1.72, 3.48) | 2.51*** (1.49, 4.22) | 2.45*** (1.46,4.13) | 2.48*** (1.47,4.19) | 2.54*** (1.51,4.28) | 2.46*** (1.45,4.15) |
| *Education* [Ref: No schooling] | | | | | | |
| Primary School | 0.99 (0.41, 2.35) | 1.61 (0.63,4.15) | 1.52 (0.59,3.93) | 1.63 (0.63,4.19) | 1.62 (0.63,4.17) | 1.54 (0.59,3.99) |
| Post-Primary/Vocational/Other | 1.31 (0.48, 3.57) | 2.09 (0.68,6.43) | 1.98 (0.64,6.14) | 2.12 (0.69,6.53) | 2.07 (0.67,6.4) | 2.01 (0.65,6.24) |
| Secondary/'A' level | 0.66 (0.27, 1.61) | 1.45 (0.53,3.98) | 1.36 (0.49,3.76) | 1.46 (0.53,4.02) | 1.45 (0.53,4.02) | 1.38 (0.5,3.83) |
| College/University | 0.83 (0.28, 2.49) | 1.02 (0.3,3.53) | 0.96 (0.28,3.32) | 1.03 (0.3,3.54) | 1.03 (0.3,3.56) | 0.97 (0.28,3.37) |
| Lower MC Use Village [Ref: Higher MC Use Village] | 0.59** (0.41, 0.83) | 0.90 (0.58,1.41) | 0.92 (0.59,1.43) | 0.90 (0.58,1.41) | 0.90 (0.58,1.4) | 0.91 (0.58,1.43) |
| Female [Ref: Male] | 0.56*** (0.41, 0.78) | 0.48*** (0.32,0.74) | 0.49*** (0.32,0.75) | 0.49** (0.32,0.75) | 0.48*** (0.31,0.73) | 0.49*** (0.32,0.75) |
| MC Use leads to health problems | 1.32*** (1.13, 1.54) | 1.13 (0.93,1.37) | 1.12 (0.92,1.35) | 1.13 (0.93,1.37) | 1.13 (0.93,1.36) | 1.12 (0.92,1.35) |
| MC Use leads to lost trust between partners | 1.19* (1.02, 1.39) | 1.09 (0.91,1.32) | 1.09 (0.9,1.32) | 1.09 (0.9,1.31) | 1.08 (0.9,1.31) | 1.08 (0.89,1.31) |
| Sexual activity is ok | 1.86*** (1.58, 2.19) | 1.56*** (1.29,1.89) | 1.57*** (1.29,1.9) | 1.57*** (1.29,1.9) | 1.57*** (1.29,1.9) | 1.57*** (1.3,1.91) |
| MC use is ok | 1.49*** (1.31,1.69) | 1.34*** (1.15,1.57) | 1.37*** (1.17,1.59) | 1.34*** (1.15,1.56) | 1.35*** (1.16,1.57) | 1.36*** (1.17,1.59) |
| Muslim [Ref: Other] | 1.34 (0.97, 1.85) | 0.99 (0.65,1.5) | 1.02 (0.67,1.55) | 1.00 (0.66,1.52) | 1.00 (0.65,1.51) | 1.03 (0.67,1.56) |
| McFadden's Pseudo $R^2$ | | 0.175 | 0.178 | 0.175 | 0.176 | 0.179 |
| Akaike Information Criterion | | 759.59 | 758.83 | 761.09 | 760.82 | 762.03 |

*p<0.05

**p<0.01

***p<0.001

Note: Continuous variables are mean-centered.

variables and individual MC use were confirmed from the bivariate analyses. They were also confirmed in the multivariable models for perceived injunctive and descriptive norms, but not for network MC use.

Our hypotheses related to significant interaction effects between normative variables and individual MC use were not supported as none were detected in Models (2)–(6). Additionally, once we controlled for perceived descriptive and injunctive norms in the multivariable models, network MC use was no longer associated with individual MC use. Model (2) showed that the two MC use belief variables (i.e. "MC use leads to health problems" and "MC use leads to lost trust between partners") were not associated with MC use, though both MC use attitude variables (i.e. "Sexual activity is ok" and "MC use is ok") remained associated with individual MC use.

**Table 5. Stratified regression results.**

| | Individual MC Use Adjusted ORs | | | |
|---|---|---|---|---|
| | Females (N = 421) | Males (N = 306) | Married Females (N = 157) | Unmarried Females (N = 264) |
| | (1) | (2) | (3) | (4) |
| Network MC Use | 2.90** (1.31,6.42) | 0.89 (0.4,1.97) | 2.51 (0.85,7.41) | 5.26* (1.34,20.69) |
| Perceived Descriptive Norms | 1.37 (0.96,1.94) | 1.28 (0.95,1.73) | 1.22 (0.73,2.06) | 1.31 (0.74,2.3) |
| Perceived Injunctive Norms | 1.21 (0.9,1.63) | 1.35 (0.98,1.86) | 1.29 (0.84,1.98) | 1.05 (0.66,1.69) |
| Age | 1.21*** (1.08,1.35) | 1.00 (0.89,1.13) | 1.16 (0.98,1.39) | 1.31*** (1.12,1.54) |
| Married [Ref: Unmarried] | 2.28* (1.16,4.49) | 1.47 (0.53,4.07) | | |
| *Education* [Ref: Primary School] | | | | |
| No Schooling | 0.46 (0.16,1.33) | 1.98 (0.1,37.33) | 0.53 (0.17,1.65) | |
| Post-Primary/Vocational/Other | 1.41 (0.57,3.46) | 1.45 (0.49,4.3) | 0.86 (0.21,3.57) | 1.36 (0.41,4.48) |
| Secondary/'A' level | 0.52 (0.24,1.12) | 1.33 (0.74,2.38) | 0.63 (0.12,3.17) | 0.52 (0.2,1.35) |
| College/University | 0.09* (0.01,0.83) | 1.60 (0.53,4.87) | 1.29 (0.1,17.11) | |
| Lower MC Use Village [Ref: Higher MC Use Village] | 0.51 (0.26, 1) | 1.35 (0.7,2.6) | 0.22** (0.09,0.58) | 1.43 (0.49,4.19) |
| MC Use leads to health problems | 0.99 (0.75,1.32) | 1.15 (0.87,1.52) | 1.20 (0.82,1.75) | 0.71 (0.41,1.24) |
| MC Use leads to lost trust between partners | 1.20 (0.9,1.59) | 1.01 (0.76,1.33) | 1.18 (0.79,1.76) | 1.36 (0.86,2.15) |
| Sexual activity is ok | 1.21 (0.86,1.7) | 1.84*** (1.43,2.36) | 1.04 (0.64,1.69) | 1.41 (0.85,2.36) |
| MC use is ok | 1.27* (1.02,1.57) | 1.47** (1.16,1.86) | 1.00 (0.74,1.37) | 1.67** (1.18,2.37) |
| Muslim [Ref: Other] | 0.89 (0.49,1.63) | 0.95 (0.51,1.77) | 1.09 (0.46,2.56) | 1.05 (0.37,2.92) |

Note: Continuous variables are not mean-centered.

*p<0.05

**p<0.01

***p<0.001

Our sex- and marriage-stratified multivariable logistic regression models are presented in **Table 5**.

The results from Table 5 illustrate several sex differences on key covariates. In particular, among female participants, we found a significant relationship between network MC use and individual MC use. This relationship did not hold for male participants. Similarly, there was a positive relationship between age and being married with individual MC use among female participants, but not among male participants. The two MC use attitude variables (i.e. "Sexual activity is ok" and "MC use is ok") were associated with individual MC use among male participants, but not among female participants. In the marriage-stratified results in models (3) and (4), network MC use was associated with individual MC use among unmarried female participants, but not among married female participants.

For our sensitivity analyses, we used a modified network MC use variable in which we imputed a value of 0 for an alter's MC use if they had missing data, -1 if the alter did not use MC, and +1 if the alter used MC (S2 Table in S1 File). We found that the results were consistent with those from our main regression results, with a slight attenuation in the relationship between individual MC and network MC use, but no difference in statistical significance. Similarly, the interaction estimates were slightly lower in magnitude compared to the main regression results. Our stratified sensitivity analysis (S3 Table in S1 File) illustrated similar estimates for perceived descriptive and injunctive norms across models compared to our main stratified regression results. However, the estimate of network MC use among unmarried women was higher in magnitude compared to our main stratified regression results.

## Discussion

This study adds to the growing body of literature on the importance of social norms around MC use in Kenya, and echoes similar recommendations to engage with larger social networks and community-level norms as a way to increase MC Use [30, 36–38]. Kenya's laws concerning adolescent MC use are generally in line with WHO recommendations, particularly around the availability and accessibility of contraceptives for adolescents [39]. However, there is a disconnect between legal protections and actual MC use, driven in large part by entrenched stigmas around female adolescent sexuality and unequal power in intimate relationships [38, 40]. The continuing impacts of COVID-19 have exacerbated these challenges. In some cases, the financial instability resulting from COVID-19 related employment shocks in Kenya may have strengthened the normative influence of referents who are against MC use [41]. It is therefore essential for future interventions and programs to measure and shift social norms around MC use. As noted by the Guttmacher-Lancet Commission, changing social and gender norms around sexual and reproductive health and rights requires the participation of multiple stakeholders, and is important to improve health, particularly among the most vulnerable to discrimination and social exclusion [42].

Our study illustrates a novel approach to estimating social influence of collective behaviors on individual MC use, that can guide future research and interventions targeting both the individual and community-levels. As public- and private-sector partners work together to achieve Kenya's FP2030 commitment to increase the voluntary use of modern contraception by anyone who wants it, social norms are an important metric that should be measured as well. While network MC use may not be a feasible metric for all settings, descriptive and injunctive norms can provide indications of progress in MC use, beyond actual reported use. Additionally, by determining the composition of referent groups (i.e. 'alters'), interventions can leverage this information in developing targeted interventions. For instance, the results from this study suggested that same-sex referents were common for both adolescent male and female participants, but especially more common adolescent female participants. This is because social norms around family planning in Kenya can reflect gender roles, attitudes, and women's autonomy, and open dialogue around these issues can actually shift social norms toward greater MC use [22]. Program planners, researchers, and policymakers should consider incorporating measures of social norms around MC use with existing indicators.

That network MC use was strongly associated with individual MC use in the bivariate models, but not in the main multivariable models, suggests that the influence of network MC use on individual MC use might be mediated by perceived norms. Thus, it may be that network MC use manifests in behaviors or communication that impacts an individual's perceived descriptive and injunctive norms, to influence behavior. Our cross-sectional study design precludes strong mediational conclusions, and thus these relationships would be better served in future studies investigating the potential mediating role of perceived norms on the network MC use-individual MC use relationship.

Additionally, our stratified results suggest that the influence of network MC use on individual MC use differs by sex, as the relationship was much larger for female participants but not for male participants. This phenomenon may reflect underlying differences between men and women for social influence effects related to individual MC use. Since the most common referents for female participants were other women, this suggests that those who affiliate with same-sex MC users are more likely to use MC themselves. Further, when the effect of network MC use was broken down by marital status, unmarried female participants were much more likely than married female participants to use MC. Future research should investigate whether affiliating with MC-using same-sex referents may blunt the stigma of MC use among unmarried females.

Our study highlighted the importance of same-sex relations as referents for male and female participants. It should be noted that the reference group in our study comprise the individuals with whom the respondent spent free time in the last year. It is possible these individuals may not be the appropriate reference group for individual MC use. However, given the significant relationship between network MC use and individual MC use for adolescent women in the sample, we believe this reference group carries some importance for family planning outcomes. Though reference groups can vary for different functions or behaviors [43, 44], it is conceivable that some reference groups may be influential for other family planning behaviors beyond MC use. Thus, future research should consider the role of same-sex referents in influencing other family planning behaviors as well.

It was surprising not to have found interaction effects between the norms variables. This does not necessarily imply that no interaction effects exist, as the study may not have been powered to detect these effects [45, 46]. Additionally, respondents provided different numbers of referents, and this may have introduced less granular estimates of group influence than would be afforded by a larger sample. Nonetheless, the significant association between network MC use and individual MC use among female participants was independent of the association between perceived norms and individual MC use. This suggests that collective behavior within a reference network can exert its own influence on individual behavior, and should be considered separately from the effects of perceived norms on behavior. Additionally, our main regression and sensitivity analysis results were similar, strengthening our confidence in the validity of the parameter estimates obtained.

This study is not without limitations. First, given the cross-sectional design of this study, we cannot establish temporal precedence of our normative variables and individual MC Use. Developmentally, we might infer that social influences and normative perceptions are formed prior to the age of sexual debut through social interaction and observation, though we cannot confirm the direction of influence from our data. Additionally, during data collection the response rate across both villages was 72%. We do not know if the individuals who did not participate significantly differ from those in our sample on key covariates or in terms of individual MC use. Therefore, it is possible that our results are reflective of selection effects. However, the omission of these individuals does not compromise the utility of our findings for future research. We have illustrated a significant relationship between social normative perceptions and individual MC use, as well as the influence of network MC use among female participants in particular. Future efforts with should seek to clarify whether the pattern between these variables holds in larger samples.

This study was unique in that it used sociocentric data to estimate the influence of collective and perceived norms on individual MC use. In this way, this study represents a bridge between social network analysis and empirical social norms analysis. It also involved collecting comparable data for both male and female participants, which is not always the case in global health surveys [47]. Consistent with recent conceptualizations of social norms [48], this study adds to the literature suggesting that social norms are multilevel phenomena and are relevant to family planning decisions among adolescents in rural Kenya.

## Supporting information

**S1 File. Questionnaire items and sensitivity analyses results.**
(DOCX)

## Author Contributions

**Conceptualization:** Jeffrey Bingenheimer, Erica Sedlander, Wolfgang Munar, Rajiv Rimal.

**Data curation:** Shaon Lahiri.

**Formal analysis:** Shaon Lahiri.

**Funding acquisition:** Wolfgang Munar.

**Investigation:** Erica Sedlander, Wolfgang Munar, Rajiv Rimal.

**Methodology:** Shaon Lahiri, Jeffrey Bingenheimer, Rajiv Rimal.

**Supervision:** Jeffrey Bingenheimer, Wolfgang Munar, Rajiv Rimal.

**Writing – original draft:** Shaon Lahiri.

**Writing – review & editing:** Shaon Lahiri, Jeffrey Bingenheimer, Erica Sedlander, Wolfgang Munar, Rajiv Rimal.

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
