## [Decision Letter · Decision Letter 0]

21 Jan 2022

PONE-D-21-26509The Role of Social Norms on Adolescent Family Planning in Rural Kilifi County, KenyaPLOS ONE

Dear Dr. Lahiri,

Thank you for submitting your manuscript to PLOS ONE. After careful consideration, we feel that it has merit but does not fully meet PLOS ONE’s publication criteria as it currently stands. Therefore, we invite you to submit a revised version of the manuscript that addresses the points raised during the review process. 

We look forward to receiving your revised manuscript.

Kind regards,

Lindsay Stark

Academic Editor

PLOS ONE

Journal Requirements:

Reviewers' comments:

Reviewer's Responses to Questions

**Comments to the Author**

1. Is the manuscript technically sound, and do the data support the conclusions?

Reviewer #1: Yes

Reviewer #2: Partly

2. Has the statistical analysis been performed appropriately and rigorously? 

Reviewer #1: Yes

Reviewer #2: Yes

3. Have the authors made all data underlying the findings in their manuscript fully available?

Reviewer #1: No

Reviewer #2: Yes

4. Is the manuscript presented in an intelligible fashion and written in standard English?

Reviewer #1: Yes

Reviewer #2: Yes

5. Review Comments to the Author

Reviewer #1: Thank you for providing me with the opportunity to review this manuscript. This manuscript explores the critical but understudied topic of social norms and contraceptive use for adolescents. Further, the authors use a clever study design to tease out the relative influence of descriptive norms, injunctive norms, and “actual” norms. I think this is a strong and useful paper and could be further strengthened by addressing the following comments.

Methods

- Can you provide a bit more detail about the original survey used to identify the two villages with lower and higher MC use? Were the two estimates of MC use (10.3% and 44.4%) derived from an adolescent population, or did the sample for those figures include adults as well?

- The Data and Study Design subsection would benefit from additional information on data collection and ethical procedures:

o Were both informed parental consent and respondent assent obtained for those under 18?

o Please include the IRB approval information

o Did you have a target sample size? Please add sample size calculations

Results

- Table 2 finds MC use to be 33% and 23% in the high and low MC use villages, respectively. Do you have a sense of why these numbers differ from the initial estimates used to select the villages (44.4% and 10.3%)? Is it because the initial estimates were derived from an adult- and not adolescent- population (this question relates to my first comment for the Methods section)?

- Apologies if I missed this somewhere in the article, but I think it would be helpful to see whether the descriptive and injunctive norms are correlated with network use. This would help to confirm whether the individuals identified as members of one’s network are comparable to “those around me” and “those close to me,” as framed in the norms questions.

-

Reviewer #2: While the topic of the paper is important, the data are unique, and the methods seem appropriate, I find the paper is written more as a report with insufficient detail and thought spent on the type of presentation that I would expect for a publication in a peer-reviewed paper. I sincerely hope that the authors re-write the paper with more care – as I think the potential is there for a very interesting paper. It is for this reason that I include the following specific comments that I hope the authors find useful.

Abstract

Overall, I find the abstract somewhat impenetrable, even for someone familiar with the approach and topic. The Purpose section is mostly methods and is missing the motivation for the research. The Methods section has insufficient details about the approach. The Results section should include some key statistics with information on the directionality of the effects. The Conclusions emphasize the findings for unmarried women, whom I assume are those “living with a partner” in this study, but the authors never make that clear. I suggest using one term or the other throughout the paper – but make it clear that unmarried doesn’t mean single. Not all readers will be familiar with the term “sociometric.”

Introduction

1. The first sentence about findings in 2020 suggests that family planning use has improved substantially from 2014 in Kenya, making it difficult for the reader to know whether they should be concerned about the 2014 statistics presented in the same paragraph. Suggest updating the 2014 statistics or re-contextualizing the information.

2. The authors should familiarize themselves with the following reference on social norms and adolescent pregnancy in Honduras (Shakya HB, Darmstadt GL, Barker KM, Weeks J, Christakis N. Social normative and social network factors associated with adolescent pregnancy: a cross-sectional study of 176 villages in rural Honduras. J Glob Health 2020;10:010706.)

3. I would expect that the aims for the research would be included in paragraph form at the end of the introduction. The presentation format for the research questions and hypotheses strikes me as a style reserved for protocols or grant proposals.

4. Regarding the research questions, I am surprised that the stratification by males/females and married/unmarried were not represented. Presumably the researchers hypothesized that there would be differences by sex and marital status given sexual double standards and gender norms in the region. If so, why not include questions to investigate these differences?

Methods

1. The authors should explain their motivation for selecting villages with a high and low prevalence MC use (e.g., to assure that they had sufficient variation in responses, perhaps?). On a related note, why did they not look at the associations stratified by village, for example, asking, whether the associations of norms wtih individual MC use differed in the high vs. low prevalence MC use villages?

2. How many people were dropped because they did not nominate a referent?

3. For the “aggregation” of two items – were these at the individual level? Is so, it would be less confusing to simply state that they were averaged. The authors refer to “aggregating” individual behaviors to enumeration areas.

4. The authors should be clear that the perceived injunctive norms were recoded, similar to the MC beliefs. Related to this – the wording for MC beliefs is negative in the tables, but I think that the finding is that “MC use does NOT lead to health problems” is associated with increased MC use. I suggest changing the table text by clarifying the reference category.

Results

1. The results are missing an introductory paragraph describing the sample numbers, response rates, and missing data information. Also, the description of results included in the text needs more detail about comparison groups (e.g., for females as compared to males) and the directionality of the association (e.g., increasing individual use with increasing age).

2. As the paper stands, I don’t find the stratification of the descriptive data in Table 2 very useful. It would be more interesting to see these data by sex and marital status. However, I am curious to know how different the injunctive norms are by village type.

3. Tables with regression results should include either SEs or CIs – my preference would be to include upper and lower 95% CIs.

4. Were other potential confounding variables considered, such as labor force participation, household or village wealth, or village size or distance to roads? If not, why not?

Discussion

1. Overall, I find the discussion weak with repetition of text from the results, little reference to how the findings fit in with what has been shown in the literature, and no discussion of broader policy or program planning implications.

2. I don’t think the findings from the sensitivity analyses about unmarried females should be emphasized in the discussion as a main finding, unless they are moved to the main paper.

3. There are a couple of comments about findings that are not presented in the text and therefore should either not be in the discussion or be added to the results:

a. “…when unmarried women come from social networks in which others (particularly same-sex others) are using contraceptives…” Results aren’t presented by whether the alter is same-sex or not (unless I missed them?).

b. “That this study was still able to detect some interactions between norms variables in the stratified analyses…” I didn’t see any results of interactions in the stratified analyses.

4. The sentence stating that “the omission of these individuals does not compromise our findings” is problematic since the omission could, in fact, result in selection bias that could result in significant but invalid findings. The authors should discuss how the missing observations are unlikely to lead to selection bias.

6. PLOS authors have the option to publish the peer review history of their article (what does this mean?). If published, this will include your full peer review and any attached files.

Reviewer #1: No

Reviewer #2: No

---

## [Author Response · Author response to Decision Letter 0]

15 Apr 2022

Response to Reviewers 

April 15, 2022

Thank you for the opportunity to revise and resubmit our manuscript. We believe that the manuscript is stronger for having addressed the thoughtful feedback from the reviewers. The following is a summary of responses to each comment from the reviewers and from PLOS ONE. Comments are in Black, and our responses are in Red (see attached Word document for colors). 

Comments from PLOS ONE

We have followed PLOS ONE’s style requirements, including those for file naming. This includes inserting line numbers, adding the title page to the manuscript, and following the file naming guidelines. 

We have included the title page at the beginning of the manuscript, with all authors and affiliations.

We have included a more complete ethics statement at the end of the Data and Study Design section in ‘Methods. We have noted the full name of the IRB which approved this study, the approval ID, and an explicit mention of informed oral consent obtained from all study participants. 

Comments from Reviewer #1

Methods

- Can you provide a bit more detail about the original survey used to identify the two villages with lower and higher MC use? Were the two estimates of MC use (10.3% and 44.4%) derived from an adolescent population, or did the sample for those figures include adults as well?

We have now clarified that that the estimates of MC use in the villages were derived from women of childbearing age (15-49 years). We have also clarified the data collection year (2018), and added a few more details about the process.

- The Data and Study Design subsection would benefit from additional information on data collection and ethical procedures:

o Were both informed parental consent and respondent assent obtained for those under 18?

Yes, we have now clarified this in the first paragraph of the “Data and Study Design” section. Individuals 18 years or older provided informed oral consent. Individuals between 15 and 17 years provided informed assent, and for them we also obtained parental consent.

o Please include the IRB approval information

IRB approval information has now been included. 

o Did you have a target sample size? Please add sample size calculations

We have now clarified in paragraph 2 of the “Data and Study Design” section that the aim was to survey 100% of the eligible men and women for the study, but the overall response rate was 72%. There were no other sample size calculations or considerations.

Results

- Table 2 finds MC use to be 33% and 23% in the high and low MC use villages, respectively. Do you have a sense of why these numbers differ from the initial estimates used to select the villages (44.4% and 10.3%)? Is it because the initial estimates were derived from an adult- and not adolescent- population (this question relates to my first comment for the Methods section)?

This is exactly right. The discrepancy is because the larger figures are for women of reproductive age (15-49), and the results in table 2 reflect the MC use for our sample (15-24). We are hopeful this is now evident from our clarification of this point in the Methods section. 

- Apologies if I missed this somewhere in the article, but I think it would be helpful to see whether the descriptive and injunctive norms are correlated with network use. This would help to confirm whether the individuals identified as members of one’s network are comparable to “those around me” and “those close to me,” as framed in the norms questions.

None of the correlations between network MC Use, descriptive norms, and injunctive norms were higher than 0.1, nor were they significant. We have now added two sentences after Table 1 to clarify this. 

Comments from Reviewer #2

Reviewer #2: While the topic of the paper is important, the data are unique, and the methods seem appropriate, I find the paper is written more as a report with insufficient detail and thought spent on the type of presentation that I would expect for a publication in a peer-reviewed paper. I sincerely hope that the authors re-write the paper with more care – as I think the potential is there for a very interesting paper. It is for this reason that I include the following specific comments that I hope the authors find useful.

Abstract

Overall, I find the abstract somewhat impenetrable, even for someone familiar with the approach and topic. The Purpose section is mostly methods and is missing the motivation for the research. The Methods section has insufficient details about the approach. The Results section should include some key statistics with information on the directionality of the effects. The Conclusions emphasize the findings for unmarried women, whom I assume are those “living with a partner” in this study, but the authors never make that clear. I suggest using one term or the other throughout the paper – but make it clear that unmarried doesn’t mean single. Not all readers will be familiar with the term “sociometric.”

We have rewritten the abstract entirely. The Purpose section now contains background information and motivation for the study. The Methods section now has much greater detail on data collection, variable creation, and analysis approach. The Results section now presents specific statistics with directionality to illustrate key results. The Conclusions section emphasizes the findings for unmarried women living with a partner. We have also clarified the “living with a partner” criterion in the Methods section of the manuscript. We have removed the term ‘sociometric’ in the Abstract, and written the methods more descriptively. 

Introduction

1. The first sentence about findings in 2020 suggests that family planning use has improved substantially from 2014 in Kenya, making it difficult for the reader to know whether they should be concerned about the 2014 statistics presented in the same paragraph. Suggest updating the 2014 statistics or re-contextualizing the information.

We have updated the statistics using data from Q1 of 2021, as reported by the Performance Monitoring and Accountability (PMA) project. We have updated a few different parts of the Introduction to reflect more recent data and findings, including how COVID-19 has impacted the rate of teen pregnancy and school dropout among girls in Kenya.

2. The authors should familiarize themselves with the following reference on social norms and adolescent pregnancy in Honduras (Shakya HB, Darmstadt GL, Barker KM, Weeks J, Christakis N. Social normative and social network factors associated with adolescent pregnancy: a cross-sectional study of 176 villages in rural Honduras. J Glob Health 2020;10:010706.)

Thank you for sharing this reference. The study by Shakya and colleagues is highly relevant to our own, and shares a similar focus on norms, and a similar methodology. We have cited this study in the first sentence of the Discussion section, as we found it fit better in that section.

3. I would expect that the aims for the research would be included in paragraph form at the end of the introduction. The presentation format for the research questions and hypotheses strikes me as a style reserved for protocols or grant proposals.

We have removed the “Research Questions and Hypotheses” section. Instead, we have described our hypotheses in narrative form in the last paragraph of the Introduction section, as recommended.

4. Regarding the research questions, I am surprised that the stratification by males/females and married/unmarried were not represented. Presumably the researchers hypothesized that there would be differences by sex and marital status given sexual double standards and gender norms in the region. If so, why not include questions to investigate these differences?

Indeed, this was the main motivation for our stratified analyses. We have now made this explicit in the last paragraph of the Introduction section, when discussing our aims and hypotheses.

Methods

1. The authors should explain their motivation for selecting villages with a high and low prevalence MC use (e.g., to assure that they had sufficient variation in responses, perhaps?). On a related note, why did they not look at the associations stratified by village, for example, asking, whether the associations of norms with individual MC use differed in the high vs. low prevalence MC use villages?

We have now explained the purpose of selecting both villages, as they represent different stages of fertility transition, in the Methods section. We chose to look only at the descriptives by village, rather than stratifying the main analyses by village, primarily due to sample size. Once we break down the sample by village, and then to further stratify on gender and marital status, would drastically reduce our statistical power to detect our relationships of interest. Given that our focus was to determine the structure of normative relationships with MC use, we opted to use a pooled sample for the regressions.

2. How many people were dropped because they did not nominate a referent?

Thirty-two individuals were dropped for not nominating a referent in the main regression models. We have now clarified this point in the second paragraph of the Methods section.

3. For the “aggregation” of two items – were these at the individual level? Is so, it would be less confusing to simply state that they were averaged. The authors refer to “aggregating” individual behaviors to enumeration areas.

We have removed “aggregate” and its variations when describing variable creation for our study. Instead, we have simply written “by averaging two items for each variable.”

4. The authors should be clear that the perceived injunctive norms were recoded, similar to the MC beliefs. Related to this – the wording for MC beliefs is negative in the tables, but I think that the finding is that “MC use does NOT lead to health problems” is associated with increased MC use. I suggest changing the table text by clarifying the reference category.

Actually, the wording in the tables is correct, and it is the response options which indicate that variables are coded such that higher values are more pro-MC. To change the wording in the tables for the MC beliefs would not reflect how the questions were asked, and would actually be the reverse of the way the variables were coded. However, we have attempted to clarify this by adding an example and making clear the direction of the response options in the last two paragraphs of the Methods section. 

Results

1. The results are missing an introductory paragraph describing the sample numbers, response rates, and missing data information. Also, the description of results included in the text needs more detail about comparison groups (e.g., for females as compared to males) and the directionality of the association (e.g., increasing individual use with increasing age).

We have added an introductory paragraph describing the sample characteristics in Table 1. Additionally, we have also added Network MC Use (with n_missing) and perceived norms to Table 1.

2. As the paper stands, I don’t find the stratification of the descriptive data in Table 2 very useful. It would be more interesting to see these data by sex and marital status. However, I am curious to know how different the injunctive norms are by village type.

We have changed Table 2 so that it is now stratified by marital status, which we believe is more useful than stratifying by village or sex. We have also now included all the norms variables in Table 1, so they can be compared across villages. 

3. Tables with regression results should include either SEs or CIs – my preference would be to include upper and lower 95% CIs.

We have now added 95% CI for all coefficients in all regression tables, including those in the supplementary file.

4. Were other potential confounding variables considered, such as labor force participation, household or village wealth, or village size or distance to roads? If not, why not? 

In general, our approach was to select a parsimonious, yet fairly representative, set of confounders to avoid overfitting the models. Village-level covariates such as size, distance to roads, and others are captured by including the village as a covariate in all models. We did not include individual income or labor force participation for two reasons. For one, the adolescent sample was young and was not characterized by high labor force participation. To account for household wealth, we selected education as a proxy, as this particular dataset did not contain an income variable. We believe that including village, age, and education, in addition to the other control variables, serve as a reasonable set of confounders.

Discussion

1. Overall, I find the discussion weak with repetition of text from the results, little reference to how the findings fit in with what has been shown in the literature, and no discussion of broader policy or program planning implications.

We have largely rewritten the Discussion section. The first three paragraphs are new content to situate the findings within the broader literature on social norms around MC use in Kenya, and also incorporates recent evidence on COVID-19 and FP decisions, and legal protections around MC use for adolescents. We have also added recommendations for measuring social norms which are absent from existing large-scale surveys like the DHS and PMA. We believe the Discussion section now addresses broader implications for programs, policy, and research in a more substantive manner.

2. I don’t think the findings from the sensitivity analyses about unmarried females should be emphasized in the discussion as a main finding, unless they are moved to the main paper.

We have removed these findings from the sensitivity analyses in the Discussion section.

3. There are a couple of comments about findings that are not presented in the text and therefore should either not be in the discussion or be added to the results:

a. “…when unmarried women come from social networks in which others (particularly same-sex others) are using contraceptives…” Results aren’t presented by whether the alter is same-sex or not (unless I missed them?).

b. “That this study was still able to detect some interactions between norms variables in the stratified analyses…” I didn’t see any results of interactions in the stratified analyses.

We have rewritten the paragraph with these findings. For point a) we have clarified that the relationship between network MC use and individual MC use was stronger for females than for males. Further, this relationship was stronger for unmarried females compared to married females. Given that same-sex referents were most commonly nominated by female participants, we suggest that future research investigate whether the stigma around MC use among unmarried females can be blunted by them affiliating with MC-using same-sex referents. For point b) we have removed this statement. 

4. The sentence stating that “the omission of these individuals does not compromise our findings” is problematic since the omission could, in fact, result in selection bias that could result in significant but invalid findings. The authors should discuss how the missing observations are unlikely to lead to selection bias.

We have now clarified this. We concede that selection bias may be at play with our sample. It is not possible to rule this out. Rather, we suggest that the omission of these individuals does not compromise the utility of our findings for future research that seeks to clarify whether these patterns hold in larger samples. We are merely suggesting the social norms are an important influence on individual MC use, and that network MC use may be especially influential for female participants in particular.

---

## [Decision Letter · Decision Letter 1]

26 May 2022

PONE-D-21-26509R1The Role of Social Norms on Adolescent Family Planning in Rural Kilifi County, KenyaPLOS ONE

Dear Dr. Lahiri,

Thank you for submitting your manuscript to PLOS ONE. After careful consideration, we feel that it has merit but does not fully meet PLOS ONE’s publication criteria as it currently stands. Therefore, we invite you to submit a revised version of the manuscript that addresses the points raised during the review process.

Both reviewers acknowledge that the manuscript is much improved with the revisions undertaken. However, a number of questions remain to be clarified from one of the reviewers - please could you address these in your next revision. 

We look forward to receiving your revised manuscript.

Kind regards,

Caroline Anita Lynch

Academic Editor

PLOS ONE

Reviewers' comments:

Reviewer's Responses to Questions

**Comments to the Author**

1. If the authors have adequately addressed your comments raised in a previous round of review and you feel that this manuscript is now acceptable for publication, you may indicate that here to bypass the “Comments to the Author” section, enter your conflict of interest statement in the “Confidential to Editor” section, and submit your "Accept" recommendation.

Reviewer #1: All comments have been addressed

Reviewer #2: (No Response)

2. Is the manuscript technically sound, and do the data support the conclusions?

Reviewer #1: Yes

Reviewer #2: Yes

3. Has the statistical analysis been performed appropriately and rigorously? 

Reviewer #1: Yes

Reviewer #2: Yes

4. Have the authors made all data underlying the findings in their manuscript fully available?

Reviewer #1: No

Reviewer #2: Yes

5. Is the manuscript presented in an intelligible fashion and written in standard English?

Reviewer #1: Yes

Reviewer #2: Yes

6. Review Comments to the Author

Reviewer #1: Thank you for sharing your revised manuscript. I feel you have adequately addressed both my and the other reviewer's comments.

Reviewer #2: The paper has been much improved. However, a number of concerns remain.

Abstract

• Methods: Unclear what the authors mean by “aggregating the proportion”

• Results: Please replace p-values with upper and lower 95% CIs for effect estimates

• Results: The “strength” of an association is determined by the magnitude of the coefficient not the p-value. Please edit accordingly.

• Results: Abbreviation MC was used without previous explanation (i.e., include (MC) after first use of modern contraception).

Introduction

• Line 149-151 – isn’t the Shakya paper an example? Another reference that is relevant is: Mejía-Guevara I, Cislaghi B, Weber A, et al. Association of collective attitudes and contraceptive practice in nine sub-Saharan African countries. J Glob Health. 2020;10(1):010705. doi:10.7189/jogh.10.010705

• Line 153 – Abbreviation MC was used without previous explanation (i.e., include (MC) after first use of modern contraception).

• Line 157 – Why do the authors hypothesize that there would be significant interaction between the 3 norm-related variables (other than the one example) in this particular context?

Methods

• What do the authors mean by higher and lower “fertility transition” for the villages?

• The content of the methods section is much better, but it needs re-organization for logical flow and additional clarity. For example, the section describing the sample is confusing with duplication of information. Referents are first described in the “data and study design” section but how they were nominated is in the subsequent section on measures. Similarly, information about the main outcome is first introduced the “data and study design” section but should be under measures with the first paragraph in this section.

• Which analyses were restricted to those married or living with a partner? Regression analyses? I see "single" in the descriptive statistics. Were singles (the majority of the sample) excluded from the regression models and why?

• Failure to nominate a referent is a criterion for exclusion from the regression analyses, but was included in the descriptive statistics. Since missing for this and individual MC use are described in the results, I think the missing info on network MC use can be removed from the methods section to avoid duplication.

• Try to keep the order of the norm-related variables the same throughout – e.g., - network MC use, descriptive norms, and perceived injunctive norms (the order they were presented in the methods).

Results

• Please clarify the sample size used in different analyses (descriptive vs. regressions). Include N’s in the tables, especially the stratified analyses.

• Why is the % of missing so high for Traditional Contraception Use among non-MC users in Table 2?

• Consider showing one star for p-value <0.1, 2 stars for <0.05, and 3 stars for <0.01. There is little to be gained by indicating p-values <0.001 over <0.01 (this is not a measure of strength but of precision). My guess is that network MC use is significant at the 10% level in the full sample. Given that the magnitude and direction of effect is consistent with the other findings, I might argue that the adjusted effect is meaningful and the wide confidence intervals may be due to insufficient power or effect modification by sex (unless the authors think the positive association is biased after adjusting for confounders).

• If, as the authors suggest in the discussion, the effect of network MC use on individual MC use is mediated through perceived norms (i.e., the perception variables are blocking part of the path), then why not include an initial model with the network variable but without the other 2 perception variables? While this approach doesn’t provide “strong” evidence of mediation – it would be suggestive and support further research.

Discussion

• What do the authors mean by “original study” vs. sample on line 443?

• I think that this study is also unique in that comparable data were collected for both females and males (see Weber, Ann M., et al. "Gender-related data missingness, imbalance and bias in global health surveys." BMJ global health 6.11 (2021): e007405).

• An additional limitation is the use of logistic regression for a common outcome. Odds ratios will overestimate the risk ratio for outcomes exceeding about 10% prevalence.

• Another limitation is that multiplicative interaction is of much less public health relevance than additive interaction (see VanderWeele, Tyler J., and Mirjam J. Knol. "A tutorial on interaction." Epidemiologic methods 3.1 (2014): 33-72.)

• The authors might be interested in the following reference for the need to measure gender norms: Weber, et.al. Gender norms and health: insights from global survey data. Lancet. 2019 Jun 15;393(10189):2455-2468. doi: 10.1016/S0140-6736(19)30765-2. Epub 2019 May 30. PMID: 31155273. Additionally, the case

Overall

• Be consistent with use of terms for participants. For example, in the paragraph starting on line 300, the authors refer to “women” and then “boys” (not “men”). I suggest using adolescent females (or girls) and adolescent males (or boys). While I assume the authors know the respondents’ sex but not their gender, they may prefer to use the terms “girls” and “boys” to emphasize their youth.

• In order for to obtain an estimate of interaction on the additive scale (and to avoid other problems with logistic regression), the authors would need to use log-binomial or Poisson models to estimate relative risk or relative rate for a common outcome. I realize that this would be a big change, so am only informing the authors for future reference in terms of the limitations of logistic regression.

• Check for typos and grammatical errors.

7. PLOS authors have the option to publish the peer review history of their article (what does this mean?). If published, this will include your full peer review and any attached files.

Reviewer #1: No

Reviewer #2: No

---

## [Author Response · Author response to Decision Letter 1]

17 Jun 2022

Response to Reviewers 

June 17, 2022

Thank you for the opportunity to revise and resubmit our manuscript. We believe that the manuscript is stronger for having addressed the additional feedback from Reviewer #2. The following is a summary of responses to each comment from Reviewer #2. Comments are in Black, and our responses are in Red. 

Comments from Reviewer #2

Reviewer #2: The paper has been much improved. However, a number of concerns remain.

Abstract

• Methods: Unclear what the authors mean by “aggregating the proportion”

This has been changed to “taking an average of an individual’s referents who use modern contraception.”

• Results: Please replace p-values with upper and lower 95% CIs for effect estimates

Done.

• Results: The “strength” of an association is determined by the magnitude of the coefficient not the p-value. Please edit accordingly.

This is a good point. We have made the corresponding changes. 

• Results: Abbreviation MC was used without previous explanation (i.e., include (MC) after first use of modern contraception).

We have changed ‘MC’ to “modern contraceptive use” in the Abstract. In the manuscript, we have introduced ‘MC’ after the first instance of “modern contraception.” 

Introduction

• Line 149-151 – isn’t the Shakya paper an example? Another reference that is relevant is: Mejía-Guevara I, Cislaghi B, Weber A, et al. Association of collective attitudes and contraceptive practice in nine sub-Saharan African countries. J Glob Health. 2020;10(1):010705. doi:10.7189/jogh.10.010705

The study by Mejía-Guevara et al. (2020) uses individual attitude items, and aggregates them to a community level for a measure of “collective attitudinal norms.” Even though the underlying methodology (of aggregation to the collective from the individual level) is similar to ours, we hesitate to rely on this for two reasons. First, as the authors concede on pg 2, attitudes are not norms, and indeed the authors use them as proxies for norms. The divergence between personal attitudes and normative perceptions can be drastic (e.g. pluralistic ignorance) and thus we believe this measure can be a potentially noisy approximation for norms. Second, ours is a measure of behavioral prevalence within a reference group, rather than a measure of attitudinal prevalence. In order to keep the two separate, we did not rely on the Mejia-Guevara et al. study, but remain open to doing so if the reviewer feels strongly that we should. 

We agree about using the Shakya study. Though they use a village-level reference group, rather than proximal reference group, to estimate collective norms, their inclusion of alters’ normative perceptions is conceptually similar to our approach. We have now included a sentence acknowledging the Shakya study in this section, and modified the concluding sentence of the paragraph to clarify that we are not aware of studies in the adolescent family planning literature in sub-Saharan Africa investigating the relationship between collective norms, operationalized as behavioral prevalence within a proximal social network, on individual behavior.

• Line 153 – Abbreviation MC was used without previous explanation (i.e., include (MC) after first use of modern contraception).

Abbreviation MC has now been placed after the first instance of the term “modern contraception” used in Sentence 2.

• Line 157 – Why do the authors hypothesize that there would be significant interaction between the 3 norm-related variables (other than the one example) in this particular context?

We have now clarified that these hypotheses stem from the Expanded Theory of Normative Social Behavior (Rimal and Yilma, 2021). This expansion suggests that collective, injunctive, and descriptive norms may operate differently for adolescent males and females, and that the interactions between normative variables can be used to understand different normative scenarios on behavior (e.g. scenarios where all norms are high, scenarios where all are low, and combinations of each). 

Methods

• What do the authors mean by higher and lower “fertility transition” for the villages?

We have now clarified that different levels of fertility transition refer to the differential rate of decline from high to lower fertility between villages. The decline was sharper in one village as compared to the other. 

• The content of the methods section is much better, but it needs re-organization for logical flow and additional clarity. For example, the section describing the sample is confusing with duplication of information. Referents are first described in the “data and study design” section but how they were nominated is in the subsequent section on measures. Similarly, information about the main outcome is first introduced the “data and study design” section but should be under measures with the first paragraph in this section.

We appreciate this suggestion, but believe that combining elements about how variables were created (from the ‘Measures’ section) with how sampling was conducted (from the “data and study design” section) would conflate the two. Rather, we prefer to explain the study design elements (such as sampling strategy and data collection process), and how the variables were created, in separate sections. Though these may appear redundant, they refer to different aspects of our study.

For example, the mention of referents in the “data and study design” section relates to the sampling design, in that trained enumerators matched egos with alters (referents) through photographs using the Trellis software. This is different from how referents are mentioned in the ‘Measures’ section, in which we created the network MC use variable by taking an average of referents’ MC use. The former refers to a data collection approach, while the latter refers to variable creation.

Similarly, if the reviewer is referring to the mention of high and low MC use villages in stating that the main outcome is introduced in the “data and study section,” this is again a question of sampling design vs measure creation. With regard to sampling, we describe how village-level MC prevalence was used a criterion for village selection. Distinctly, individual MC use was described as the main outcome in the ‘Measures’ section. The former refers to sampling and the latter refers to a different variable used for our statistical models. We hope the reviewer agrees about the underlying rationale – to keep separate the description of the study design from that of the measures. 

• Which analyses were restricted to those married or living with a partner? Regression analyses? I see "single" in the descriptive statistics. Were singles (the majority of the sample) excluded from the regression models and why?

Thank you for catching that. We have removed the sentence stating that the analysis was restricted to those married or living with a partner. The analysis included those who were unmarried (single), who were not cohabiting with a partner, as well as those who were married or cohabiting with a partner.

• Failure to nominate a referent is a criterion for exclusion from the regression analyses, but was included in the descriptive statistics. Since missing for this and individual MC use are described in the results, I think the missing info on network MC use can be removed from the methods section to avoid duplication.

We have removed this information from the methods section.

• Try to keep the order of the norm-related variables the same throughout – e.g., - network MC use, descriptive norms, and perceived injunctive norms (the order they were presented in the methods).

We have now ensured the norms variables are presented in this order in the narrative and for the main effects in all tables.

Results

• Please clarify the sample size used in different analyses (descriptive vs. regressions). Include N’s in the tables, especially the stratified analyses.

The Ns are now present in all tables in the Headers.

• Why is the % of missing so high for Traditional Contraception Use among non-MC users in Table 2?

The denominator for the missing in Traditional Contraception Use among non-MC users is the full sample (n = 763), most of whom were MC users. For clarity, we have now removed this row. 

• Consider showing one star for p-value <0.1, 2 stars for <0.05, and 3 stars for <0.01. There is little to be gained by indicating p-values <0.001 over <0.01 (this is not a measure of strength but of precision). My guess is that network MC use is significant at the 10% level in the full sample. Given that the magnitude and direction of effect is consistent with the other findings, I might argue that the adjusted effect is meaningful and the wide confidence intervals may be due to insufficient power or effect modification by sex (unless the authors think the positive association is biased after adjusting for confounders).

We discussed this issue internally and came to the conclusion that, as our alpha threshold of 0.05 was determined a priori, we should retain our star levels as is – with the conventional *for p<0.05, ** for p<0.01, and *** for p<0.001. This is also consistent with most published literature in our field. As we have included 95% CIs for coefficients, readers will be able to see if certain estimates have come close to the threshold. We hope the reviewer agrees with this decision.

• If, as the authors suggest in the discussion, the effect of network MC use on individual MC use is mediated through perceived norms (i.e., the perception variables are blocking part of the path), then why not include an initial model with the network variable but without the other 2 perception variables? While this approach doesn’t provide “strong” evidence of mediation – it would be suggestive and support further research.

We appreciate the suggestion to include a model without the perceived norms variables. We have been careful to include both explanatory and exploratory models we had specified a priori, and wish to avoid running additional models for this study. Further, three factors make us think that such a model might not be necessary: 1) it is likely that we may not have sufficient statistical power to detect a mediational effect, if it exists, 2) a significant network MC use coefficient in a model without perceived norms may reflect an inflated Type I error due to multiple hypothesis tests from additional models, and 3) our cross-sectional study design precludes the establishment of temporal precedence. Thus, we would not be able to argue convincingly that network MC use influences perceived norms, or the other way around. For these reasons, we believe a more robust treatment of mediation is warranted in future studies, that are well-powered a priori to detect potential mediation effects. This is described in the Discussion.

Discussion

• What do the authors mean by “original study” vs. sample on line 443?

We refer to the study for which the data were collected, that included individuals aged 15-49 (our study is restricted to individuals aged 15-24). To clarify, we have removed the term “original study” and rephrased two sentences to refer to the individuals who did not participate during data collection.

• I think that this study is also unique in that comparable data were collected for both females and males (see Weber, Ann M., et al. "Gender-related data missingness, imbalance and bias in global health surveys." BMJ global health 6.11 (2021): e007405).

We appreciate this point, and have added it to our Discussion section. 

• An additional limitation is the use of logistic regression for a common outcome. Odds ratios will overestimate the risk ratio for outcomes exceeding about 10% prevalence.

While we do not dispute that odds ratios overestimate risk ratios when outcomes are not rare (i.e. when they exceed 10%), we do not see the use of logistic regression as a general limitation that compromises the utility of our findings. Odds ratios are a commonly reported metric in the social norms and family planning literature, and our use of them is in line with similar studies that have used logistic regression to report odds ratios, and have not mentioned this as a general limitation (e.g. Mejía-Guevara et al., (2020), Sedlander & Rimal (2019; 2021), Wegs et al., (2016), Komasawa et al., (2020), etc). 

• Another limitation is that multiplicative interaction is of much less public health relevance than additive interaction (see VanderWeele, Tyler J., and Mirjam J. Knol. "A tutorial on interaction." Epidemiologic methods 3.1 (2014): 33-72.)

We very much appreciate this point and the reference. Our decision to use multiplicative interactions was to conduct analyses in line with the Expanded Theory of Normative Social Behavior (Rimal & Yilma, 2021) which uses multiplicative interactions. This is not to say, that additive interactions would not be a useful statistical complement to the theory’s propositions, but we chose our analyses guided by this theory, for which currently only multiplicative interactions have been applied. Further, the multiplicative interactions are tied to our hypothesis, particularly in how they would be falsified (i.e. by non-significant interaction terms, ostensibly from multiplicative interactions). However, this suggestion is very useful, and we will certainly be considering how to incorporate additive interactions in future work. In future interaction models, we will also consider how to specify a priori how hypotheses would be falsified if the results from multiplicative and additive interactions are different. 

• The authors might be interested in the following reference for the need to measure gender norms: Weber, et.al. Gender norms and health: insights from global survey data. Lancet. 2019 Jun 15;393(10189):2455-2468. doi: 10.1016/S0140-6736(19)30765-2. Epub 2019 May 30. PMID: 31155273. Additionally, the case

We appreciate the reference and will consult it for future research. The reviewer’s comment cuts off in the last sentence.

Overall

• Be consistent with use of terms for participants. For example, in the paragraph starting on line 300, the authors refer to “women” and then “boys” (not “men”). I suggest using adolescent females (or girls) and adolescent males (or boys). While I assume the authors know the respondents’ sex but not their gender, they may prefer to use the terms “girls” and “boys” to emphasize their youth.

We appreciate this comment and have now used the terms male and female participants throughout.

• In order for to obtain an estimate of interaction on the additive scale (and to avoid other problems with logistic regression), the authors would need to use log-binomial or Poisson models to estimate relative risk or relative rate for a common outcome. I realize that this would be a big change, so am only informing the authors for future reference in terms of the limitations of logistic regression.

Thank you, this is very useful to know for future interaction models. Hopefully, the reviewer agrees with the point above (about sticking with the multiplicative model to align with extant literature), which would obviate the need to incorporate that point in this paper.

• Check for typos and grammatical errors.

Done.

---

## [Decision Letter · Decision Letter 2]

26 Aug 2022

PONE-D-21-26509R2The Role of Social Norms on Adolescent Family Planning in Rural Kilifi County, KenyaPLOS ONE

Dear Dr. Lahiri,

Thank you for submitting your manuscript to PLOS ONE. After careful consideration, we feel that it has merit but does not fully meet PLOS ONE’s publication criteria as it currently stands. Therefore, we invite you to submit a revised version of the manuscript that addresses the points raised during the review process.

The reviewers have a small number of outstanding comments that should be addressed.

We look forward to receiving your revised manuscript.

Kind regards,

Jamie Royle

Staff Editor

PLOS ONE

Journal Requirements:

Reviewers' comments:

Reviewer's Responses to Questions

**Comments to the Author**

1. If the authors have adequately addressed your comments raised in a previous round of review and you feel that this manuscript is now acceptable for publication, you may indicate that here to bypass the “Comments to the Author” section, enter your conflict of interest statement in the “Confidential to Editor” section, and submit your "Accept" recommendation.

Reviewer #1: All comments have been addressed

Reviewer #2: All comments have been addressed

2. Is the manuscript technically sound, and do the data support the conclusions?

Reviewer #1: Yes

Reviewer #2: Yes

3. Has the statistical analysis been performed appropriately and rigorously? 

Reviewer #1: Yes

Reviewer #2: Yes

4. Have the authors made all data underlying the findings in their manuscript fully available?

Reviewer #1: No

Reviewer #2: Yes

5. Is the manuscript presented in an intelligible fashion and written in standard English?

Reviewer #1: Yes

Reviewer #2: Yes

6. Review Comments to the Author

Reviewer #1: I suggested accepting this manuscript during the previous round of revisions and have no additional comments.

Reviewer #2: I wish to thank the authors for their consideration of my comments, which were addressed. I have two very minor suggestions, which I do not need to review:

* In the abstract, I still find the statement unclear: "...we estimated group-level normative influence by taking an average of an individual’s referents who use modern contraception." Could use text in author's response to a previous comment: "...by taking an average of referents’ modern contraception use."

* Line 168 at the end of the introduction: change "gender" to "sex" in :"we also stratified our analyses by gender and marital status"

7. PLOS authors have the option to publish the peer review history of their article (what does this mean?). If published, this will include your full peer review and any attached files.

Reviewer #1: No

Reviewer #2: No

---

## [Author Response · Author response to Decision Letter 2]

26 Aug 2022

Response to Reviewers

August 26, 2022

Thank you for the opportunity to revise and resubmit our manuscript. We have made the two minor revisions requested by Reviewer #2. Comments from Reviewer #2 are in Black, and our responses are in Red. 

Comments from Reviewer #2

 I wish to thank the authors for their consideration of my comments, which were addressed. I have two very minor suggestions, which I do not need to review:

* In the abstract, I still find the statement unclear: "...we estimated group-level normative influence by taking an average of an individual’s referents who use modern contraception." Could use text in author's response to a previous comment: "...by taking an average of referents’ modern contraception use."

Done.

* Line 168 at the end of the introduction: change "gender" to "sex" in :"we also stratified our analyses by gender and marital status"

Done.

---

## [Editor Report · Decision Letter 3]

26 Sep 2022

The Role of Social Norms on Adolescent Family Planning in Rural Kilifi County, Kenya

PONE-D-21-26509R3

Dear Dr. Lahiri,

We’re pleased to inform you that your manuscript has been judged scientifically suitable for publication and will be formally accepted for publication once it meets all outstanding technical requirements.

Kind regards,

Dario Ummarino, PhD

Senior Editor

PLOS ONE

---

## [Editor Report · Acceptance letter]

24 Jan 2023

PONE-D-21-26509R3 

The Role of Social Norms on Adolescent Family Planning in Rural Kilifi County, Kenya 

Dear Dr. Lahiri:

I'm pleased to inform you that your manuscript has been deemed suitable for publication in PLOS ONE. Congratulations! Your manuscript is now with our production department. 

Kind regards, 

on behalf of

Jamie Royle 

%CORR_ED_EDITOR_ROLE%

PLOS ONE